# Impact of infectious diseases consultation on the management of *Staphylococcus aureus* bacteraemia in children

Rebecca B Saunderson,[1] Theodore Gouliouris,[1,2,3] Edward J Cartwright,[1,3] Emma J Nickerson,[4] Sani H Aliyu,[2,4] D Roddy O'Donnell,[5] Wilf Kelsall,[5] D Limmathurotsakul,[6] Sharon J Peacock,[1,2,3,7] M Estée Török[1,2,3]

For numbered affiliations see end of article.

**Correspondence to**
Dr M Estée Török;
et317@medschl.cam.ac.uk

## ABSTRACT

**Objectives:** Infectious diseases consultation (IDC) in adults with *Staphylococcus aureus* bacteraemia (SAB) has been shown to improve management and outcome. The aim of this study was to evaluate the impact of IDC on the management of SAB in children.

**Study design:** Observational cohort study of children with SAB.

**Setting:** Cambridge University Hospitals National Health Service (NHS) Foundation Trust, a large acute NHS Trust in the UK.

**Participants:** All children with SAB admitted to the Cambridge University Hospitals NHS Foundation Trust between 16 July 2006 and 31 December 2012.

**Methods:** Children with SAB between 2006 and 31 October 2009 were managed by routine clinical care (pre-IDC group) and data were collected retrospectively by case notes review. An IDC service for SAB was introduced in November 2009. All children with SAB were reviewed regularly and data were collected prospectively (IDC group) until 31 December 2012. Baseline characteristics, quality metrics and outcome were compared between the pre-IDC group and IDC group.

**Results:** There were 66 episodes of SAB in 63 children—28 patients (30 episodes) in the pre-IDC group, and 35 patients (36 episodes) in the IDC group. The median age was 3.4 years (IQR 0.2–10.7 years). Patients in the IDC group were more likely to have echocardiography performed, a removable focus of infection identified and to receive a longer course of intravenous antimicrobial therapy. There were no differences in total duration of antibiotic therapy, duration of hospital admission or outcome at 30 or 90 days following onset of SAB.

**Conclusions:** IDC resulted in improvements in the investigation and management of SAB in children.

## INTRODUCTION

*Staphylococcus aureus* bacteraemia (SAB) is a serious infection that leads to significant morbidity and mortality in adults and children.[1 2]

## Strengths and limitations of this study

- This is the first study to examine the impact of introduction of infectious diseases consultation (IDC) on the management of *Staphylococcus aureus* bacteraemia (SAB) in children.
- We found that IDC was associated with an improvement in investigation and management of SAB, but there was no difference in mortality between the pre-IDC and IDC groups.
- The main limitation of the study was the size of the study population, which may explain the lack of mortality benefit.
- The study was conducted in a tertiary referral centre, where clinical management is likely to have been good prior to introduction of the IDC, and may not be generalisable to other settings.

*S. aureus* causes significant disease in the paediatric population, occurring in 1.5% of all neonatal intensive care unit (ICU) admissions,[3] and 6/100 000 children older than 1 year of age.[4] In neonates, SAB is almost always hospital acquired, and is frequently due to intravascular catheter (IVC)-associated infections.[3–6] The majority of non-neonatal cases of SAB are community-acquired; those that are hospital-acquired infections are usually IVC associated.[7 8]

Identified risk factors for the development of SAB in the paediatric population include having a pre-existing medical condition, prolonged hospitalisation, the presence of an IVC and HIV infection.[1 3–6 9 10] Mortality from SAB in the adult population is about 30%.[11] Mortality rates in the paediatric population tend to be lower, but can be up to 15% in neonates and/or children with comorbidities.[1 5 9 12 13] Given that SAB causes a substantial burden of disease in the paediatric population, strategies to improve

management, prevent the complications of SAB and reduce mortality are a clinical priority.

The impact of infectious disease consultation (IDC) in adults with SAB has been extensively studied.[14–23] IDC has been associated with improved adherence to guidelines, including appropriate and targeted investigation, optimal duration of antibiotic therapy and a reduction in complicated infection, morbidity and mortality.[14 16–18 22 24–27] In contrast, the impact of an IDC on the management and outcomes of SAB in children has not previously been evaluated. The aim of this study was to determine the effects of routine IDC on the investigation, management and outcome of children with SAB.

## MATERIALS AND METHODS
### Study setting and participants
Cambridge University Hospitals National Health Service Foundation Trust (CUH) is a tertiary referral centre for paediatrics in the east of England. The paediatric service has a 22-bed medical and surgical ward, a 17-bed paediatric haematology and oncology ward, an 11-bed paediatric ICU and high dependency unit (caring for children aged from 0 to 16 years) and a 12-bed surgical and medical ward for children aged up to 3 years. The Rosie Hospital, the on-site mother and baby hospital, has a 17-cot neonatal ICU and a 10-cot Special Care Baby Unit.

### Study design
We conducted an observational cohort study of all children with SAB admitted to CUH between 16 July 2006 and 31 December 2012. In November 2009, an IDC service for all patients with SAB was established at CUH. Data were collected from 2006 to 2009 by a retrospective review of the medical records, and prospectively thereafter during the IDC service. Patients with blood cultures that were considered to be contaminants (afebrile with no clinical evidence of infection) or with polymicrobial blood cultures were excluded from the analysis.

### Microbiological investigation
Blood cultures were collected and incubated at 37°C for 5 days using the BacT/Alert 3D system (bioMérieux, Basingstoke, UK). Blood cultures that flagged positive were examined by microscopy and presumptively identified as *S. aureus* using a thermostable nuclease test.[28] Colonies of *S. aureus* were identified by routine methods after a further overnight incubation. Identification of *S. aureus* was performed using a commercial latex agglutination test (Staphaurex, Oxoid Ltd, Basingstoke, UK) until 2011 and then using matrix-assisted laser desorption ionisation time-of-flight mass spectroscopy (Bruker Daltonik, Bremen, Germany). Antibiotic susceptibilities were determined using disc diffusion testing, according to the British Society for Antimicrobial Chemotherapy standards.[29] Throughout the study period a clinical microbiologist provided telephone advice to the clinical

team for all patients with SAB, and attended weekly ward rounds on the paediatric oncology ward and paediatric ICU.

### Study procedures
Prior to November 2009, all patients with *S. aureus* bacteraemia were managed by their primary clinical care team, with telephone advice from the microbiologists. From November 2009, all patients with SAB were reviewed by an infectious disease Specialist Registrar or Consultant, following presumptive identification of *S. aureus* in blood cultures. The assessment included clinical history to determine symptoms of infection, and physical examination to determine possible foci of infection. Patients underwent clinical review daily by their primary care team and at least weekly by the IDC team during their inpatient stay. Demographic, clinical and microbiology data were collected using a standard case record form, and entered into an electronic database.

An IVC was considered to be the focus of infection if there was evidence of inflammation at the catheter exit site and/or a vascular catheter tip culture positive for *S. aureus*, without clinical evidence of another source of bacteraemia.[30] Thrombophlebitis was diagnosed when there was clinical evidence of infection and inflammation along a blood vessel or when ultrasound or other imaging confirmed the presence of intravascular thrombosis in the setting of suspected infection. Bone and joint infections were defined according to the USA Centers for Disease Control and Prevention criteria.[31] The lung was considered to be the source of infection when there was clinical, radiological and/or microbiological evidence of pulmonary infection. Soft tissue infection was considered to be the source of the bacteraemia if the clinical signs of a known or suspected soft tissue infection predated or were present at the time of bacteraemia.[14] A deep tissue abscess was defined by radiological imaging criteria. Infective endocarditis (IE) was diagnosed according to the modified Duke criteria.[32 33]

A SAB episode was defined as being greater than or equal to 14 days from a previous episode, in the absence of persistent bacteraemia or focus of infection. A secondary site of infection was defined as a site of infection separate from the primary site of infection that was not present at the time of the initial examination. Healthcare-associated bloodstream infection was defined according to previously published criteria.[34] Hospital-acquired infection was defined according to the USA Centers for Disease Control and Prevention.[31] Community-acquired infections were defined as those patients with a positive blood culture taken at or within 48 h of admission who did not meet criteria for healthcare-associated bloodstream infection.[34] Patients were classified as having uncomplicated SAB if blood cultures were negative 2–4 days after the initial blood culture was positive, if they had defervesced at 72 h, if

there was no evidence of metastatic disease or endocarditis or if they had a catheter-related infection.[35]

Appropriate antimicrobial therapy was defined as therapy to which the isolate was determined to be susceptible by antimicrobial disc susceptibility testing. The duration of therapy was the length of time that a patient received antibiotics to which the isolate was susceptible. An underlying medical condition was defined as any chronic medical condition that was present at the time of bacteraemia. Serum C reactive protein, blood white cell counts and platelet counts were measured on the day of, or within 48 h postbacteraemia. Duration of hospital admission and outcome at 30 and 90 days postbacteraemia was recorded for all patients.

### Treatment recommendations

Antimicrobial treatment recommendations were provided for all children with SAB, based on existing evidence on the management of SAB in adults.[14 36–39] These included removal of a removable focus of infection,[14] performing repeat blood cultures at 48–96 h,[36] performing a transthoracic echocardiogram, performing radiological imaging of suspected deep foci of infection, treating uncomplicated infection with 14 days of intravenous antibiotics,[37] treating complicated infections with a minimum of 28 days of intravenous antibiotics,[38] and using β-lactam therapy as the mainstay of treatment for methicillin-susceptible S. aureus.[39]

### STATISTICAL ANALYSIS

Data were analysed using STATA V.12 (StataCorp, College Station, Texas, USA). Categorical variables were analysed using Fisher's exact test and reported as the number and per cent. Continuous variables were compared using the Mann Whitney U test and reported as the median and IQR. Mortality was analysed per patient (ie, only the first bacteraemia episode was analysed).

### ETHICS STATEMENT

Written informed consent from participants was not required as the study was conducted as a service evaluation.

### RESULTS
### Patient characteristics

Between July 2006 and December 2012, 71 children had one or more blood cultures that were positive for S. aureus. Sixty-three children (66 episodes) were included in the study. Five children (six episodes) were excluded because of polymicrobial bacteraemia and three patients (three episodes) were excluded because the cultures were considered to be contaminants. Thus, 28 patients (30 episodes) were included in the pre-IDC group, and 35 patients (36 episodes) in the IDC group. The study schema is summarised in figure 1. Four of 30 episodes (13.3%) received an IDC before the service was implemented in 2009, and 34 of 36 episodes (94.4%) received an IDC after the service was implemented in 2009.

The baseline characteristics of the two groups were similar (table 1). The clinical features for SAB were likewise similar, apart from an increased proportion of IVC-related infections in the IDC group (61.3% vs 26.7%, p<0.01; table 2). A higher proportion of patients had an unidentified focus of infection in the pre-IDC group compared with the IDC group (23.3% vs 5.6%, p=0.07). Risk factors for SAB were also similar apart from an increased frequency of prosthetic material in the IDC group (72.2% vs 46.7%, p=0.04; table 3).

### Clinical management

A service evaluation of the IDC service was conducted, the results of which are summarised in table 4. In the IDC group, 34/36 episodes had an infectious diseases review, with a median time to review of 2 days (range 1–4 days). Patients in the IDC group were more likely to have transthoracic echocardiography performed (80.6% vs 33.3%, p<0.01). They were also more likely to have a removable focus of infection identified (43.9% vs 23.3%, p<0.01), although there was no difference between the two groups in the likelihood of removal, or the time to removal. In the IDC group, two patients did not have their IVC removed, despite the recommendation to do so, because of concerns about difficulty in re-establishing vascular access. There was no difference in the number of repeat blood cultures performed between groups.

### Antimicrobial therapy

There was no difference between the two groups in the time taken to initiate appropriate antimicrobial therapy. Patients in the IDC group were more likely to receive a longer duration of intravenous antimicrobial therapy (18 vs 13.5 days, p=0.04), although there was no difference in total duration of therapy (intravenous and oral) between the two groups. In patients with complicated SAB, the duration of intravenous antibiotic therapy was longer in the IDC group (22 vs 14 days, p=0.02), although there was no difference in total duration of antibiotic therapy (intravenous and oral) between the two groups. Patients in the IDC group were more likely to receive a longer duration of intravenous therapy if their repeat blood culture result was positive (p<0.01). In patients with uncomplicated SAB. there was no difference between groups in the duration of intravenous antibiotics, or the total duration of antibiotic therapy. In terms of compliance with recommended standards for duration of therapy, patients in the IDC group were more likely to meet these standards compared with patients in the non-IDC group, both for complicated SAB (42.1% vs 13.3%, p=0.13) and uncomplicated SAB (68.4% vs 46.7%, p=0.14). There was no difference in the proportion of patients receiving β-lactam therapy for methicillin-sensitive S. aureus bacteraemia between the two groups.

**Figure 1** Study schema of paediatric patients with *Staphylococcus aureus* bacteraemia.

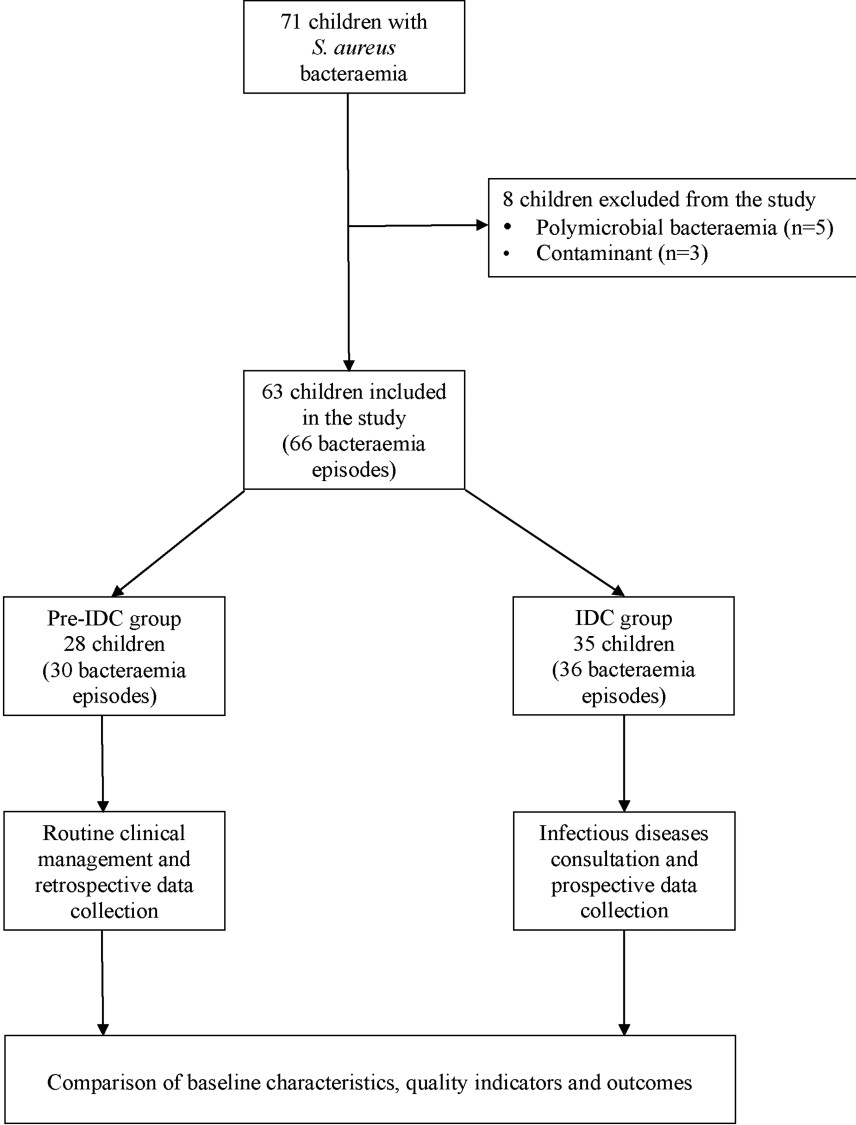

## Outcome of SAB

The duration of hospital admission was similar in the pre-IDC and IDC groups, and in those with uncomplicated and complicated SAB (table 4). SAB was recorded in the discharge summary in the majority of patients in both groups. Four secondary infections were diagnosed, three in the IDC group and one in the pre-IDC group. In the IDC group, the secondary infections were tricuspid valve endocarditis (in a very low birthweight neonate with patent ductus arteriosus), pneumonia and osteomyelitis, respectively. One child in the non-IDC group developed osteomyelitis. There were three cases of recurrent bacteraemia within 90 days—two in the pre-IDC group and one in the IDC group. Three children died within 30 days of SAB, all in the IDC group, giving an overall 30-day mortality rate of 4.8%. One death occurred in a child with metastatic cancer that was not attributed to SAB. The other two deaths were deemed attributable to SAB, as blood cultures were positive at the time of death. One patient was a neonate and died prior to IDC, and the second case had cerebellar

atrophy, developmental delay and was receiving total parenteral nutrition. The 90-day mortality rate was 7.9%. Two patients died between 30 and 90 days post-SAB, both in the pre-IDC group. One patient had metastatic cancer, and the other had complex congenital heart disease.

## DISCUSSION

To the best of our knowledge, this is the first study to systematically examine the impact of introduction of an IDC service on the management of SAB in children. We compared the clinical features, management and outcomes of all children presenting with SAB to our hospital between July 2006 and December 2012, before and after introduction of an IDC service. The main findings of the study were that patients in the IDC group were more likely to have echocardiography performed, a removable focus of infection identified and to receive a longer course of intravenous antibiotic therapy. These findings concur with those from previous studies of IDC

**Table 1**  Baseline characteristics of children with *Staphylococcus aureus* bacteraemia included in the study

| Baseline variable | Pre-IDC group N=28 patients (%) | IDC group N=35 patients (%) | Combined N=63 patients (%) |
|---|---|---|---|
| Male | 20 (71.4) | 20 (57.1) | 40 (63.5) |
| Female | 8 (28.6) | 15 (42.9) | 23 (36.5) |
| Median age in years (IQR) | 4.3 (0.2–9.4) | 3.4 (0.2–12.2) | 3.4 (0.2–10.7) |
| Neonates | 6 (20) | 9 (25.0) | 15 (23.8) |
| Prematurity | 4 (66.7) | 9 (100.0) | 13 (86.7) |
| Median age in days (IQR) | 11 (7–47) | 26 (23–38) | 25 (14–47) |
| Median birth weight in grams (IQR) | 1117 (630–3535) | 820 (755–1120) | 830 (710–1330) |
| Congenital heart disease | 5 (17.9) | 8 (22.9) | 13 (20.6) |
| Chronic pulmonary disease | 3 (10.7) | 5 (14.3) | 8 (12.7) |
| Liver disease | 0 | 1 (2.9) | 1 (1.6) |
| Malignancy | 7 (25.0) | 9 (25.7) | 16 (25.4) |
| Metastatic cancer | 2 (7.1) | 1 (2.9) | 3 (4.8) |
| Neurological condition | 4 (14.3) | 13 (37.1) | 17 (27.0) |
| Diabetes mellitus | 0 | 1 (2.9) | 1 (1.6) |
| Skin condition | 3 (10.7) | 1 (2.9) | 4 (6.4) |
| Atopic dermatitis | 3 (100.0) | | 3 (75.0) |
| Immunosuppression | 6 (21.4) | 8 (22.9) | 14 (22.2) |
| Corticosteroid therapy | 2 | 3 | 5 |
| Antineoplastic | 5 | 5 | 10 |
| Neutropenia | 2 | 0 | 2 |
| | N=30 episodes (%) | N=36 episodes (%) | N=66 episodes (%) |
| Mode of acquisition | | | |
| Community-acquired | 11 (36.7) | 10 (27.8) | 21 (31.8) |
| Healthcare-associated | 9 (30.0) | 7 (19.4) | 16 (24.2) |
| Hospital-acquired | 10 (33.3) | 19 (52.8) | 29 (43.9) |
| Duration of symptoms of bacteraemia (hours) | | | |
| 0–24 | 21 (70.0) | 18 (50.0) | 39 (59.1) |
| 25–72 | 1 (3.3) | 6 (16.7) | 7 (10.6) |
| >72 | 7 (23.3) | 10 (27.8) | 17 (25.8) |
| Unknown | 1 (3.3) | 2 (5.6) | 3 (4.6) |
| Organism | | | |
| MSSA | 28 (93.3) | 33 (91.7) | 61 (92.4) |
| MRSA | 2 (6.7) | 3 (8.3) | 5 (7.6) |
| C reactive protein (nmol/L) | 333 (114–890) | 448 (181–1081) | 390 (133 –1005) |
| White cell count ($10^9$/L) | 8.8 (5.8–15.6) | 10.1 (6.9–18.4) | 9.6 (6.3–16.5) |
| Neutrophils ($10^9$/L) | 5.4 (3.6–9) | 6.3 (4.9–11.8) | 5.7 (3.6–11.8) |
| Platelets ($10^9$/L) | 277 (151–374) | 213 (101–270) | 224 (140–301) |

IDC, infectious diseases consultation; MSSA, methicillin-susceptible *Staphylococcus aureus*; MRSA, methicillin-resistant *Staphylococcus aureus*.

conducted in adults with SAB, and reflect current best practice.

Follow-up blood cultures have been recommended in adults, as prolonged bacteraemia is a predictor of complicated infection and poorer outcome in SAB.[36] As a result, prolonged intravenous antibiotic therapy is recommended in patients with positive repeat blood cultures. We found that children who had a positive repeat blood culture were more likely to receive a longer course of intravenous antibiotics if they were in the IDC group.

Echocardiography was performed in a higher proportion of children in the IDC group compared with the pre-IDC group. The rates of IE in children with SAB are reported to be between 0% and 20%, which is similar to rates reported in the adult population.[1 3 6 8 12 40–42] An American study by Valente *et al*[40] diagnosed IE in 20%

of children with SAB (~12% of whom had confirmed IE). Children with underlying congenital heart disease had a higher prevalence of confirmed or probable IE compared with those who had structurally normal hearts (53% vs 3%) and patients with definite IE had multiple positive blood cultures. Mortality was higher in patients with endocarditis compared with those without (40% vs 12%). Another study from South Africa reported an IE rate of 11% in children with SAB.[12] Risk factors for the development of IE in children include congenital heart disease, a central IVC and persistently positive blood cultures after 24 h.[40 41] In the UK, there are no published guidelines on the use of echocardiography in children with SAB. The Infectious Diseases Society of America guidelines for meticillin-resistant *S. aureus* (MRSA) bacteraemia recommend performing echocardiography in

**Table 2** Clinical features of *Staphylococcus aureus* bacteraemia in children included in the study

| Focus of infection at time of bacteraemia | Pre-IDC group N=30 episodes (%) | IDC group N=36 episodes (%) | Combined N=66 episodes (%) | p Value |
|---|---|---|---|---|
| Unknown focus | 7 (23.3) | 2 (5.6) | 9 (13.6) | NS |
| Intravascular catheter | 8 (26.7) | 22 (61.3) | 30 (45.5) | 0.013 |
| Culture confirmed | 5 (62.5) | 11 (50.0) | 16 (53.3) | NS |
| Thrombophlebitis | 0 | 2 (5.7) | 2 (3.0) | NS |
| Bone/joint infection | 8 (26.7) | 8 (22.2) | 16 (24.2) | NS |
| Culture confirmed | 4 (50.0) | 2 (25.0) | 6 (37.5) | NS |
| Lung | 1 (3.3) | 1 (2.8) | 2 (3.0) | NS |
| Culture confirmed | 0 | 1 (100.0) | 1 (50) | NS |
| Skin and soft tissue | 7 (23.3) | 9 (25.0) | 16 (24.2) | NS |
| Culture confirmed | 4 (57.1) | 8 (88.9) | 12 (75.0) | NS |
| Deep tissue abscess | 2 (6.7) | 1 (2.8) | 3 (4.6) | NS |
| Culture confirmed | 2 (100.0) | 0 | 2 (66.7) | NS |
| Other focus | 2 (6.7) | 4 (11.1) | 6 (9.1) | NS |
| Defervescence at 72 h | | | | |
| Yes | 18 (60.0) | 20 (55.6) | 38 (57.6) | NS |
| No | 11 (36.7) | 14 (38.9) | 25 (37.9) | |
| Unknown | 1 (3.3) | 2 (5.6) | 3 (4.6) | |

IDC, infectious diseases consultation; NS, non-significant.

children with congenital heart disease, those with bacteraemia duration greater than 2 days or those with other clinical findings suggestive of endocarditis.[35] In our study, the one child who developed tricuspid valve endocarditis was a very low birthweight premature neonate with a patent ductus arteriosus, an IVC-related infection and persistent bacteraemia. Our findings concur with these guidelines, and support the use of a risk-based strategy for the use of echocardiography in children with SAB.

We found that a higher proportion of children had an IVC-related infection and/or a removable focus of infection in the IDC group compared with the pre-IDC group, although neither of these differences was statistically significant. It is possible that IVC was used more during the IDC period compared with the pre-IDC period. Although removable foci of infection were more frequently identified and removed in the IDC group

compared with the pre-IDC group, the median time to removal was slightly longer (3 vs 2 days). In some cases, this was related to practical difficulties in removing the focus, such as re-establishing vascular access in neonates. Conversely, there were fewer patients with an unidentified focus of infection in the IDC group compared with the pre-IDC group. These findings indicate that the introduction of specialist IDC service improved the rate of diagnosis of the focus of infection in children with SAB, suggesting that the consult service was beneficial.

The frequency of MRSA bacteraemia was only 7.6% in our study cohort. Possible explanations for this are that MRSA bacteraemia is less common in children than in adults, and that MRSA bacteraemia rates have significantly declined in the UK since 2006. In contrast, in other countries such as the USA, the incidence of MRSA bloodstream infections has been higher than in the UK, but has recently declined. In a large retrospective study

**Table 3** Risk factors for *Staphylococcus aureus* bacteraemia in children included in the study

| Risk factor for SAB | Pre-IDC group N=30 episodes (%) | IDC group N=36 episodes (%) | All children N=66 (episodes) | p Value |
|---|---|---|---|---|
| Age <1 year | 11 (36.7) | 14 (38.9) | 25 (37.9) | NS |
| Underlying medical condition | 16 (53.3) | 24 (66.7) | 40 (60.6) | NS |
| Duration in hospital in days (IQR) Prior to bacteraemia* | 11.5 (7.0–21.0) | 19.0 (12.0–37.0) | 16.0 (8.0–24.0) | NS |
| Prosthetic material | 14 (46.7) | 26 (72.2) | 40 (60.6) | 0.04 |
| Intravascular line | 13 (43.3) | 23 (63.9) | 36 (54.6) | NS |
| Endotracheal tube | 0 | 4 | 4 | NS |
| Other | 1 | 6 | 8 | NS |
| corticosteroid therapy | 2 (6.7) | 3 (8.3) | 5 (7.6) | NS |
| Surgery within previous 30 days | 4 (13.3) | 4 (11.1) | 8 (12.1) | NS |

*Hospital-acquired infection only.
IDC, infectious disease consultation; NS, non-significant.

**Table 4** Comparison of the management and outcome of *Staphylococcus aureus* bacteraemia in children, preintroduction and postintroduction of an infectious disease consult service

| Quality indicator | Pre-IDC group N=30 episodes (%) | IDC group N=36 episodes (%) | p Value |
|---|---|---|---|
| Median time to infectious diseases review in days (IQR) | N=4<br>3.5 (0.5–21.5) | N=34<br>2.0 (1.0–4.0) | NS |
| Repeat blood culture performed | 26 (86.7) | 32 (88.9) | NS |
| Time to repeat blood culture (h) | | | |
| 0–48 | 20 | 18 | NS |
| 48–96 | 8 | 13 | NS |
| >96 | 1 | 1 | NS |
| Repeat blood culture positive (h) | | | |
| 0–48 | 6 | 6 | NS |
| 48–96 | 3 | 4 | NS |
| >96 | 0 | 0 | NS |
| Echocardiogram performed | | | |
| Yes | 10 (33.3) | 29 (80.6) | 0.0001 |
| No | 20 (66.7) | 7 (19.4) | |
| β-Lactam therapy | 27 (90.0) | 34 (94.4) | NS |
| MSSA | 25 (92.6) | 33 (97.1) | |
| Removable focus of infection | 7 (23.3) | 22 (43.9) | 0.003 |
| Focus removed | 6 (85.7) | 21 (95.5) | NS |
| Median time to removal in days (IQR) | 2.0 (2.0–2.0) | 3.0 (1.0–18.0) | NS |
| Median time to appropriate antibiotics in days (IQR) | 0.0 (0) | 0.0 (0) | NS |
| Median duration of intravenous antibiotics in days (IQR) | 13.5 (7.0–21.0) | 18.0 (15.0–29.0) | 0.035 |
| Median duration of intravenous and/or oral antibiotics in days (IQR) | 20.5 (16.0–42) | 19.0 (15.0–29.5) | NS |
| **Complicated infection** | **N=15 episodes<br>Days (IQR)** | **N=19 episodes<br>Days (IQR)** | |
| Median duration of intravenous antibiotics | 14.0 (6.0–21.0) | 22 (15.0–39.0) | 0.02 |
| Median duration of intravenous or oral antibiotics | 19.0 (17.0–43.0) | 27.0 (16.0–39.0) | NS |
| Median duration of intravenous antibiotics if repeat blood culture positive | 13.0 (6.0–14.0) | 19.0 (15.0–27.0) | 0.007 |
| Met standard recommendation of 28 days intravenous antibiotics (%) | 2 (13.3) | 8 (42.1) | NS |
| **Uncomplicated infection** | **N=15 episodes<br>Days (IQR)** | **N=17 episodes<br>Days (IQR)** | |
| Median duration of intravenous antibiotics | 13.0 (7.0–22.0) | 15.0 (14.0–21.0) | NS |
| Median duration of intravenous or oral antibiotics (IQR) | 22.0 (14.0–32.0) | 18.0 (14.0–29.0) | NS |
| Met standard recommendation of 14 days (%) | 7 (46.7) | 13 (68.4) | NS |
| **Outcomes** | **N=30 episodes<br>Days (IQR)** | **N=36 episodes<br>Days (IQR)** | |
| Median duration of hospital admission | | | |
| Total | 14.0 (6.0–42.0) | 16.5 (7.5–58) | NS |
| Complicated | 20.0 (6.0–49.0) | 25.0 (8.0–88.0) | NS |
| Uncomplicated | 7.0 (3.0–22.0) | 11.0 (6.0–36.0) | NS |
| SAB recorded in discharge summary | 24 (80.0) | 29 (82.9) | NS |
| Secondary infection detected | 1 | 3 | NS |
| Outcomes 30-day post-SAB | | | |
| Death | 0 | 3 | NS |
| Recurrence | 0 | 0 | NS |
| Outcomes 30–90 days post-SAB | | | |
| Death | 2 | 0 | NS |
| Recurrence | 2 | 1 | NS |

Mortality was analysed per patient (only the first episode was analysed).
IDC, infectious diseases consultation; NS, non-significant; SAB, *Staphylococcus aureus* bacteraemia.

of over 57 000 hospitalised children with *S. aureus* infections, 51% had MRSA and 61% had MRSA skin and soft tissue infections. The incidence of skin and soft tissue infections, pneumonia, osteomyelitis and bacteraemia increased over time but overall mortality was low (1%). Thus, the findings of our study may not be generalisable to other settings where the epidemiology and outcomes of MRSA bacteraemia are different.

In terms of duration of antimicrobial therapy, we found that patients in the IDC group received longer courses of intravenous antibiotics in complicated infection, compared with patients in the pre-IDC group. This concurs with findings from studies in adults with SAB, and suggests that specialist infectious diseases review may be beneficial in ensuring that clinical management recommendations, such as the length of intravenous antimicrobial therapy, are applied. There were, however, no differences in mortality observed between the pre-IDC and IDC groups. The most likely explanation for this was the small study population, combined with a low mortality rate, which meant that a large number of patients would be required to demonstrate even a small difference in mortality. Furthermore, the overall duration of antibiotic therapy was similar in the two groups. It may be that total duration of antibiotic therapy is more important than the route of administration, provided that adequate concentrations are achieved in the bloodstream. Indeed, studies to examine this very question, comparing short versus long courses of intravenous antibiotics in SAB in adults, are ongoing. Finally, although a removable focus of infection was identified more frequently in the IDC group, the likelihood of removal and the time to removal did not differ; this may also explain the lack of difference in outcome between the two groups.

We acknowledge several limitations to our study. The study population was small and the differences in diagnosis and management that we observed in the two groups did not translate into differences in outcome, for reasons discussed above. The retrospective data collection during the pre-IDC period (2006–2009) carries a risk of incomplete recording of data and potential bias. There were, however, no differences in baseline characteristics between the two groups in terms of age, gender, underlying comorbidities or focus of infection. The only exception was a higher frequency of IVC-related infections in the IDC group, which may have been under-recorded in the pre-IDC period and/or diagnosed more frequently in the IDC period.

In conclusion, we found that introduction of IDC for paediatric SAB resulted in improvements in management and a more consistent approach to care across the paediatric service. These findings concur with those of previous studies of IDC in adults with SAB. Our findings support the use of a targeted approach to echocardiography in SAB in children, particularly for patients with risk factors for complicated disease. Despite improvements in the investigation and clinical management, we did not find any differences in the development of secondary infections, recurrent bacteraemia or death between the two groups. The most likely explanation for this is the small study population and larger prospective studies are required to validate our findings and to determine the optimal strategies for investigation and management of paediatric SAB.

**Author affiliations**
[1]Department of Medicine, University of Cambridge, Cambridge, UK
[2]Department of Microbiology, Cambridge University Hospitals NHS Foundation Trust, Cambridge, UK
[3]Public Health England, Clinical Microbiology and Public Health Laboratory, Cambridge, UK
[4]Department of Infectious Diseases, Cambridge University Hospitals NHS Foundation Trust, Cambridge, UK
[5]Department of Paediatrics, Cambridge University Hospitals NHS Foundation Trust, Cambridge, UK
[6]Mahidol Oxford Research Unit, Mahidol University, Bangkok, Thailand
[7]Wellcome Trust Sanger Institute, Hinxton, UK

**Contributors** RBS, TG, EJN, SHA and DRO were part of the infectious diseases consultation service; they collected the data and contributed to the writing of the manuscript. WK was part of the clinical care team and contributed to the writing of the manuscript. SJP and MET conceived and supervised the study and contributed to the writing of the manuscript. All authors approved the final manuscript.

**Funding** This work was supported by grants from the UK Clinical Research Collaboration (UKCRC) Translational Infection Research Initiative (TIRI); the Medical Research Council (G1000803), with contributions from the Biotechnology and Biological Sciences Research Council, the National Institute for Health Research (NIHR) on behalf of the UK Department of Health, and the Chief Scientist of the Scottish Government Health Directorate; the Public Health England; and the NIHR Cambridge Biomedical Research Centre. MET is a Clinician Scientist Fellow funded by the Academy of Medical Sciences at the Health Foundation.

**Competing interests** None.

**Ethics approval** The study protocol received approval from the University of Cambridge Human Biology Research Ethics Committee, and from the CUH Research and Development Department.

**Provenance and peer review** Not commissioned; externally peer reviewed.

**Data sharing statement** No additional data are available.

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
