## [Reviewer comments · BMJ Open]

Some articles will have been accepted based in part or entirely on reviews undertaken for other BMJ Group journals. These will be reproduced where possible.

ARTICLE DETAILS

TITLE (PROVISIONAL)	Impact of infectious diseases consultation on the management of S. aureus bacteraemia in children
AUTHORS	Saunderson, Rebecca; Gouliouris, Theodore; Cartwright, Edward; Nickerson, Emma; Aliyu, Sani; O'Donnell, Roddy; Kelsall, Wilf; Limmathurotsakul, Direk; Peacock, Sharon; Torok, Estee

VERSION 1 - REVIEW

REVIEWER	Owen Robinson Royal Perth Hospital Perth, WA Australia
REVIEW RETURNED	26-Feb-2014

GENERAL COMMENTS	Major comments: In this manuscript, researcher from Cambridge, UK try to assess the impact of infectious diseases consult (IDC) on children with S aureus bacteremia (SAB). The manuscript is well written and easy to follow. There are however some methodological issue which should be addressed: First, a patient could be enrolled more than once which is a problem as some of the endpoints such as mortality are not independent... In short, a patient enrolled more than once by definition cannot die in the first episode. I suggest excluding secondary/tertiary episodes. The other major point for review/comment is that 4/28 (14.3%) patients (? Number of episodes) have had an IDC in the pre-2009 group and 34/35 (97.1% [in the manuscript, end of fist paragraph of results is stated 94.4% which is 34/36 episodes and not patient]) have not had an IDC in the post-2009 group. Given the small sample size, that's close to 10% of episode which have been analysed as having an IDC when they did not or vice versa. I suggest that the analysis be done without these cases and if this does not affect the overall results, adding a comment that tis is the case. Minor comments:
--

	 ▪ 3rd paragraph introduction: “ Has been extensively.” The word studied is missing ▪ 1st paragraph material and methods: “ medicine ward” suggest switching to medical ward (same as 2lines above) ▪ Study design, last sentence: “contaminated blood cultures.... were excluded”. Please define as 3 episodes (4.2%) were excluded and the basis that they were contaminants which is much higher than figures usually reported. ▪ Microbiology investigations: Please specify if there has been any change in the laboratory methods over the 6 years of the study. Introduction of a PCR based identification/mecA on whole blood has been shown to reduce the time to appropriate therapy ▪ The study is underpowered to had p values with more than 2 decimals. Please adjust. ▪ One of the main findings is that in the IDC group, a removable focus of infection was more often found. However in the IDC group, there were more intravascular catheter (61.3% vs 26.7%). It is therefore not entirely correct to say that a focus is more often found, as by definition, no catheter mean no catheter related infection...Please comment. ▪ The data suggest that IDC has increased the duration of IV therapy but not the total duration of therapy. I'm not that we have enough evidence that this may be beneficial as stated in discussion, but it is certainly more in keeping with guidelines. ▪ In the limitations paragraph, second line, the negation was omitted: “... that we observed in the two groups did [add NOT] translate into difference in outcome. ▪ In the conclusion, the authors say: “These findings..... support the use of echocardiography in SAB in children”. The data presented do not support this conclusion.
--	---

REVIEWER	Federico Laham, MD Arnold Palmer Hospital Medical Center Medical Director, Pediatric Infectious Diseases Orlando, Florida USA
REVIEW RETURNED	03-Mar-2014

GENERAL COMMENTS	This is a 3-year-pre/3-year-post intervention study conducted in a relatively small pediatric center in Cambridge, UK. Clinical outcomes and management of children with S.aureus bacteremia were compared regarding the presence of guidance by infectious diseases subspecialists. The topic is not novel although pediatric-specific studies are lacking. Few proof-reading issues: ----- Page 6, line 50. There is a missing word: "The impact of ... has been extensively [blank]." Studied? Page 9, line 30. Centers for Disease Control AND PREVENTION.
--

(Add)

To address:

Page 15, line 34. The authors should remind the reader that the difference of patients with unidentified focus was not statistically significant ($p > 0.05$)

Page 15, line 41. The paragraph is misleading. The results indicate that during the IDC period more patients had their SAB due to IVC-related infections, which in particular would not have require an IDC to be able to diagnose them. It does not follow then that an IDC would be beneficial. On the contrary, a non-apparent source, such as deep seated infections (pneumonia, osteomyelitis, endocarditis, etc) would have been more difficult to identify for the non-ID trained physician and thus may have resulted in the "benefit" as implied in the paragraph, but this was not the case of how the clinical presentations were distributed. Overall, the authors may comment that the distribution of disease presentation may have diluted the effects of the intervention.

Page 16, first paragraph, lines 3-7. Please reword the sentence to add clarity.

PAge 16, line 21. Delete: "and should also be conducted in the paediatric population"

PAge 17, line 5. The statement "and support the use of echocardiography in SAB in children" is not supported by the study results. Remove.

Questions

1. How was the IDC composed?

2. The authors should comment in the discussion the different types of support treating physicians had in the management of SAB prior to the implementation of IDC. For instance, Infectious diseases management recommendations may have come in a form of curbside consultations (thus missed from the retrospective data gathering), pharmacists, microbiologists, etc.

3. Intervention. It is implied that the IDC took place on day 0 and then weekly. Does this mean no interval patient assessments were done?

4. There is abundant literature about the greater clinical impact and adverse outcomes associated with bacteremia due to Methicillin-resistant Staphylococcus Aureus (MRSA). It can be hypothesized that a greater benefit may be obtained from an IDC in SAB due to MRSA. In the present study, only 7% of SAB were due to MRSA. The authors should comment about this, and also indicate that results may not be generalizable to other settings (in the US, MRSA bacteremia can be commonly as high as 50%).

5. As the authors indicate, the utility of routine TTE in uncomplicated SAB has not been established (even by the current UK guidelines). Therefore, compliance with this practice should not be used as an indicator of quality. Nevertheless, the manuscript suggest that failure to obtain a TTE is in detriment to the patient.

	6. Why took longer to remove IVC in the IDC group? The authors do not indicate what was the adherence, at least perceived, with the IDC recommendations. 7. For the 2 deaths attributable to SAB in the IDC group, was there any possible benefit added by the IDC with the patient management overall? (for example, an ID subspecialist may have readily recognized the importance of a specific intervention on a patient and steered the managing physician in that direction).
--	---

VERSION 1 – AUTHOR RESPONSE

Reviewer 1

Major comments:

- In this manuscript, researchers from Cambridge, UK try to assess the impact of infectious diseases consult (IDC) on children with *S aureus* bacteremia (SAB). The manuscript is well written and easy to follow. There are however some methodological issues which should be addressed:

- First, a patient could be enrolled more than once which is a problem as some of the endpoints such as mortality are not independent. In short, a patient enrolled more than once by definition cannot die in the first episode. I suggest excluding secondary/tertiary episodes.

We thank the reviewer for this comment. We have amended the analysis and only analysed outcome for the first bacteraemia episode.

- The other major point for review/comment is that 4/28 (14.3%) patients (? Number of episodes) have had an IDC in the pre-2009 group and 34/35 (97.1% [in the manuscript, end of first paragraph of results is stated 94.4% which is 34/36 episodes and not patient]) have not had an IDC in the post-2009 group. Given the small sample size, that's close to 10% of episodes which have been analysed as having an IDC when they did not or vice versa. I suggest that the analysis be done without these cases and if this does not affect the overall results, adding a comment that this is the case.

We thank the reviewer for this comment. We have adjusted the analysis so that it refers to bacteraemia episodes throughout. We do not think that it is reasonable to exclude cases in the pre-IDC group that were referred for and received IDC as these patients are likely to have had more serious / complicated disease. Excluding them would bias the results and may result in overestimation of the effects in the IDC group.

- Minor comments:

- o Introduction 3rd paragraph: "has as been extensively." The word studied is missing. This has been corrected.

- o Materials and methods 1st paragraph: "medicine ward" suggest switching to medical ward (same as 2 lines above). This has been corrected.

- o Study design, last sentence: "contaminated blood cultures.... were excluded". Please define as 3 episodes (4.2%) were excluded and the basis that they were contaminants which is much higher than figures usually reported. We have defined contamination more clearly in the manuscript. However, we do not think that our contamination rates are particularly high given that a previous paediatric study (J Infect. 2006 Dec;53 (6):387-93) reported contamination rates of 9%

- o Microbiology investigations: Please specify if there has been any change in the laboratory methods over the 6 years of the study. Introduction of a PCR based identification/mecA on whole blood has been shown to reduce the time to appropriate therapy. The methods of identification of *S. aureus* and MRSA did not change during the study period in a way that would influence time to appropriate therapy. The only change was the introduction of MALDI TOF MS towards the end of the study, but this is no faster than the thermostable nuclease test.

- The study is underpowered to had p values with more than 2 decimals. Please adjust.

We have corrected this so that all values with a p value <0.01 have been reported as such

- One of the main findings is that in the IDC group, a removable focus of infection was more often found. However in the IDC group, there were more intravascular catheter (61.3% vs 26.7%). It is therefore not entirely correct to say that a focus is more often found, as by definition, no catheter mean no catheter related infection. Please comment. We have clarified this to indicate that the proportion of IVC-related infections was higher in the IDC group. This may be related to increased use of IVCs during the IDC period.
- The data suggest that IDC has increased the duration of IV therapy but not the total duration of therapy. I'm not sure that we have enough evidence that this may be beneficial as stated in discussion, but it is certainly more in keeping with guidelines. We agree that the total duration of therapy may be more important than the route of administration and have addressed this in detail in the discussion. Indeed this question is being addressed in ongoing clinical trials of *S. aureus* bacteraemia.
- In the limitations paragraph, second line, the negation was omitted: "... that we observed in the two groups did [add NOT] translate into difference in outcome. We have corrected this
- In the conclusion, the authors say: "These findings support the use of echocardiography in SAB in children". The data presented do not support this conclusion. We have amended this to say that our findings support a risk-based strategy for echocardiography for children with SAB.

Reviewer 2

This is a 3-year-pre/3-year-post intervention study conducted in a relatively small pediatric center in Cambridge, UK. Clinical outcomes and management of children with *S.aureus* bacteremia were compared regarding the presence of guidance by infectious diseases subspecialists. The topic is not novel although pediatric-specific studies are lacking.

Proof reading issues

- Page 6, line 50. There is a missing word: "The impact of ... has been extensively [blank]." Studied? This has been corrected.
- Page 9, line 30. Centers for Disease Control AND PREVENTION. (Add) This has been corrected.

Issues to address

- Page 15, line 34. The authors should remind the reader that the difference of patients with unidentified focus was not statistically significant ($p > 0.05$). We have amended this.
- Page 15, line 41. The paragraph is misleading. The results indicate that during the IDC period more patients had their SAB due to IVC-related infections, which in particular would not have require an IDC to be able to diagnose them. It does not follow then that an IDC would be beneficial. On the contrary, a non-apparent source, such as deep seated infections (pneumonia, osteomyelitis, endocarditis, etc) would have been more difficult to identify for the non-ID trained physician and thus may have resulted in the "benefit" as implied in the paragraph, but this was not the case of how the clinical presentations were distributed. Overall, the authors may comment that the distribution of disease presentation may have diluted the effects of the intervention. We disagree with the reviewer on this point. We found that IVC-related bacteraemia was identified more frequently in the IDC group, suggesting that bedside clinical review was helpful in diagnosing even simple IVC-related infections.
- Page 16, first paragraph, lines 3-7. Please reword the sentence to add clarity. We have amended this.
- Page 16, line 21. Delete: "and should also be conducted in the paediatric population". We have amended this.
- Page 17, line 5. The statement "and support the use of echocardiography in SAB in children" is not supported by the study results. Remove. We have amended this to suggest a risk-based strategy for the use of echocardiography in children.

Questions

- How was the IDC composed? This has been clarified and expanded as follows: Prior to November 2009 all patients with *S. aureus* bacteraemia were managed by their primary clinical care team, with telephone advice from the microbiologists. From November 2009, all patients with SAB were reviewed by an infectious diseases Specialist Registrar or Consultant, within one working day of presumptive identification of *S. aureus* in blood cultures. The assessment included clinical history to determine symptoms of infection, and physical examination to determine possible foci of infection. Patients underwent clinical review daily by their primary care team and at least weekly by the IDC team during their inpatient stay.
- The authors should comment in the discussion the different types of support treating physicians had in the management of SAB prior to the implementation of IDC. For instance, Infectious diseases management recommendations may have come in a form of curbside consultations (thus missed from the retrospective data gathering), pharmacists, microbiologists, etc. This has been clarified in the text – see above.
- Intervention. It is implied that the IDC took place on day 0 and then weekly. Does this mean no interval patient assessments were done? This has been clarified in the text – see above
- There is abundant literature about the greater clinical impact and adverse outcomes associated with bacteremia due to Methicillin-resistant *Staphylococcus Aureus* (MRSA). It can be hypothesized that a greater benefit may be obtained from an IDC in SAB due to MRSA. In the present study, only 7% of SAB were due to MRSA. The authors should comment about this, and also indicate that results may not be generalizable to other settings (in the US, MRSA bacteremia can be commonly as high as 50%). We have added some information about differences in the epidemiology of MRSA bacteraemia between the UK and other countries and acknowledged that our findings may not be generalizable to other settings.
- As the authors indicate, the utility of routine TTE in uncomplicated SAB has not been established (even by the current UK guidelines). Therefore, compliance with this practice should not be used as an indicator of quality. Nevertheless, the manuscript suggests that failure to obtain a TTE is in detriment to the patient. We disagree with the reviewer on this point. The use of echocardiography in the investigation of SAB in adults is well established and recommended by existing guidelines, although recent data suggest that it may not be necessary in low risk patients (Kaasch A et al., Clin Infect Dis. 2011 Jul 1;53(1):1-9). In children the US guidelines recommend echocardiography in children with MRSA bacteraemia considered to be at high risk of endocarditis.
- Why took longer to remove IVC in the IDC group? The authors do not indicate what was the adherence, at least perceived, with the IDC recommendations. We have clarified this as follows: In the IDC group two patients did not have their IVC removed, despite the recommendation to do so, because of concerns about difficulty in re-establishing vascular access.
For the 2 deaths attributable to SAB in the IDC group, was there any possible benefit added by the IDC with the patient management overall? (for example, an ID subspecialist may have readily recognized the importance of a specific intervention on a patient and steered the managing physician in that direction). The patients were managed by the primary clinical care team and received regular review by the IDC service. The patients who died were neonates with multiple co-morbidities or had malignant disease and we consider it unlikely that the IDC service could have prevented their deaths.

I trust that our responses will be satisfactory and I look forward to hearing from you soon.

VERSION 2 – REVIEW

REVIEWER	Federico Laham, MD Arnold Palmer Hospital for Children Orlando, FL United States
REVIEW RETURNED	07-Jun-2014

- The reviewer completed the checklist but made no further comments.